# The Catholic Church's Formula for Adaptation to Modernity and Contemporary Models of Secularism

Piotr Musiewicz 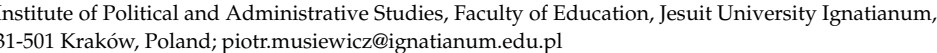

Institute of Political and Administrative Studies, Faculty of Education, Jesuit University Ignatianum, 31-501 Kraków, Poland; piotr.musiewicz@ignatianum.edu.pl

**Abstract:** The aim of this paper is to assess the Catholic Church's formula for adaptation to Modernity from the perspectives of contemporary models of Secularism. For this purpose, it will use the typology of Jacques Berlinerblau, with five models of Secularism (Separationism, Disestablishment, Laïcité, Accommodationism, and Atheistic Secularism). For some of these models, the Church's formula will be found—to different extents—agreeable, while other models will regard it as hardly acceptable. The assessment will proceed by defining Modernity, outlining the Catholic Church's formula for adaptation to modern ideas, and discussing five different models of Secularism.

**Keywords:** Modernity; modernization; Secularism; models of Secularism; Catholic Church





## 1. Introduction

Up to the early nineteenth century, most states were connected with one particular religion, and, at the core, their models of church and state relations remained similar. They shared some form of establishment, though differing in particular instances in the scope of influence of the state on the church and vice versa or in the level of religious freedom for those professing other faiths. The state's support for a particular church was, however, reduced—gradually or by revolution—due to one of the crucial tenets of Modernity: accommodation of different religions in one society (Bruce and Voas 2023). As a result, the main Modern paradigm of church and state relations, especially in the Western world, shifted from establishment to some form of separation of church and state. This idea, in its various forms, is a crucial component of contemporary Secularism; therefore, all contemporary models of Secularism have embraced the underlying idea of separation.

More traditional secularization theories indicate an ongoing diminution of the role of religion due to the progress of modernization. Casanova argued that the separation of church and state and progressive modernization provoked further differentiation of spheres; this remained the core of the theory of secularization. However, he added, this did not cause religion to disappear (Casanova 1994, p. 7). Even if the debate on secularization theories has currently reached a "dead end" (Moniz 2023), the question of the role of religion in contemporary secular states seems to be still open and vital, and might be considered in many aspects without discussing general secularization theories.

Certainly, contemporary models of Secularism are visible signs of Modernity. Typologies and models of Secularism have been widely researched; we can find a variety of these in contemporary scholarship (Saeed 2017). Another question important to Modernity— whether and to what extent the Catholic Church underwent modernization—seems to have been quite widely researched, even if the answers may differ slightly (Hastings 1991; McGreevy 2022; Faggioli 2020). However, scholarship has not yet examined the relation of the two above issues to one another. The questions of particular interest in this paper are, therefore, the following: Do the realities of Secularism and the Catholic Church have something in common? Does the Catholic formula of adaptation to Modernity resemble contemporary models of Secularism? Most of all, how do contemporary models of Secularism perceive the Catholic Church's formula?

**2. Modernity and the Catholic Church's Formula of Adaptation**

*2.1. Modernity*

After Shmuel Eisenstadt proved there is no one pattern of Modernity and that there exist and are born many quite different Modernities (Eisenstadt 2000), an attempt to propose any universally adequate and accepted definition of Modernity seems futile. Therefore, in this paper there can only be proposed a definition of Modernity eligible for its purposes and—hopefully—some new findings. Certainly, the definition will be vulnerable to arguments pointing out that the Western model of Modernity is not the only legitimate model in the world and that there are many other cultural programs different than Judeo-Christian. However, acknowledging it, the more Western or "static" definition of Modernity presented below might perhaps be justified. Modernity in the paper is analyzed with reference to the Catholic Church's formula of adaptation to it, and the center of the Church—from where the official ideas are issued—remains embedded in or at least surrounded and influenced by the Western cultural context (Judeo-Christian cultural program), and therefore Western, especially European, version of Modernity. Therefore, in this paper, when speaking of the Catholic Church's formula of adaptation to Modernity, "Western Modernity" is meant (defined below). Certainly, research concerning other patterns of Modernities and the topic might be made elsewhere, but the paper is confined to making new findings on this particular pattern of Modernity.

The definition of Modernity is, then, to some extent, derived from classical and re-vised (sociological) theories of modernization. Modernity is to be understood as a set of fundamental political, intellectual, and religious ideas prevailing in the West, significantly different from those prevailing in traditional societies of the Medieval era: as a set of ideas focusing on the autonomy of man (Eisenstadt 2000, p. 5). Therefore, the definition will not discuss whether transformation to Modernity is a coherent, linear process functioning in all societies (a version of Marx, Durkheim, Weber, and later Fukuyama) or an unpredictable process that might lead to different results (a version of Eisenstadt) (Feldhay 2022; Moniz 2023). For the purpose of this paper, it is enough to indicate the following crucial ideas of (Western) Modernity: individualization (the individual becomes as or more important than the community), sovereignty of the people (as opposed to the monarch), democratization, rationalization (looking for rational rather than religious explanations of phenomena), autonomy of science from religion, a plurality of worldviews, and a focus on individual economic development. Implementation of these values in Western societies led to func-tional differentiation, most clearly seen in the separation of church and state, the separation of science and religion, and the individualization and privatization of faith (Inglehart and Welzel 2005).

*2.2. The Catholic Church's Formula of Adaptation*

Modernization in the West has not been confined to states and societies; it has also affected churches and religions. The Catholic Church is no exception here, although she directly resisted modernization up to the latter half of the twentieth century. In 1864, as an appendix to the *Quanta Cura* encyclical, Pius IX issued the *Syllabus of Errors,* in which he articulated the Church's disagreement with—and condemnation of—individualization, rationalization, liberalism, the idea of the separation of church and state, religious freedom, limiting the Pope's power to the spiritual realm, and some other modern trends. In addition, the idea that "The Roman Pontiff can, and ought to, reconcile himself, and come to terms with progress, liberalism and modern civilization" was described as one of the errors of modern liberalism (Pius IX 1864, p. 80).

Since Vatican Council II, the above stance is no longer valid as the Catholic Church has adapted to Modernity, at least in her own formula—this process was called *aggiornamento*. However, it should be noted, this adaptation has not fully resembled the modernization of Western societies. The latter underwent revolutionary, or sometimes evolutionary, changes towards democracy: protecting individuality, separating church and state, and diminishing the role of religion in public life. In many cases, these changes were so fundamental that they

shook the foundations of social and political orders. As to the Catholic Church, although the discussion about the continuity or discontinuity of Catholic ideas at Vatican II is still open (O'Collins 2012), it can be stated that the general approach of the Church to Modernity has changed, as she saw the need to adopt Modern ideas. This need might have resulted either from general modernizing trends in the West, the experience of twentieth-century totalitarianism (Chappel 2018), or both. In any case, this meant change, and discontinuity in many ideas; however, it did not automatically alter the foundations of Church order as in the cases of states and societies. The Catholic Church at Vatican II did not accept all Modern ideas. Particularly, it denied full rationalization as in its teaching the Revelation—meaning the Scriptures and the Catholic Church's Tradition—remains and should remain a crucial source of worldview and assessing reality. Furthermore, it adopted only some elements of democratization—the role of bishops presiding together as Council or regional conference decisively increased, but still the laity, as a whole, plays no role in deciding local parish or pastoral matters, nor in electing vicars, bishops, or popes. Finally, individualization undertaken by the Catholic Church did not become as overwhelming as in Liberalism, as it still focuses also on issues of communities and its duties: families, nations, the whole Church, and global society. From a broader perspective, the Modern ideas were combined or balanced with more permanent Church principles. However, even with the above qualification, the Catholic Church's adaptation to Modernity has been considerable and remains a significant phenomenon worthy of extensive research, especially from secular perspectives.

*2.3. Particular Modern Ideas Adopted by the Catholic Church*

Concepts of popular sovereignty and democracy were implemented into Catholic doctrine at Vatican II and afterwards. It is remarkable that in *Lumen Gentium*, a crucial document on the nature of the Church, the chapter about the People of God is placed before the chapter on hierarchy (Paul VI 1964, pp. 9, 18). The focus on the common dignity of priests and laymen is also a clear sign of democratization and a mark of equality. Regarding bishops, a new concept of collegiality is frequently used, elevating their mutual relations and encouraging them to preside together—a visible departure from papal or monarchical absolutism. The Church herself is to be also a sign of unity among all men (Paul VI 1964, p. 1); this again emphasizes human relations. There has been an acceptance of diversity in discipline, liturgy, theology, and spirituality, and of local Churches; this might be regarded as the introduction of pluralism within the Church. A departure from exclusivity is also seen in the focus on common bonds with other Christians and people of different faiths (hitherto close contacts of this sort were neither encouraged nor prohibited): for example, in the addressing of the Church's social messages to all people of good will (not only to members of the Church) and in the possibility of salvation for non-Catholics (McBrien 1991).

Generally, in *Gaudium et Spes*, this is presented not so much as the normative teaching of the Church but rather the current human situation in present contexts—which were not previously seen as serious points of reference. The achievements of science are no longer seen as a threat. Democracy has become clearly accepted. Natural law categories, formerly dominant in discussing social, political, and economic issues, have been replaced by liberal and human rights categories. Indeed, the United Nations Declaration of Human Rights of 1948 has started to be integrated into Catholic teaching, along with the need for wider international cooperation. The language of documents has refrained from condemnation of other views and become more based on proposing and encouraging. Personalism has been widely introduced, with a close focus on man and his dignity. Finally, a "rightful independence of earthly affairs" (Paul VI 1965b, p. 36) is clearly stated, marking, perhaps unwillingly, a direction towards functional differentiation in some other areas (McDonagh 1991).

Human dignity has become a crucial, fundamental category. Valuing individuality has resulted in statements in favor of freedom of inquiry and communication: the index

of prohibited books has been terminated, as well as the necessity for publications to have "imprimatur" to be safe to be read by Catholics. Religious freedom has been acknowledged so that religious beliefs can be embraced only as the free act of an individual and not as a consequence of any extrinsic coercion. The Catholic Church has unambiguously accepted the plurality of worldviews and now has to rely on persuasion. That, in turn, has transformed the Church's view on church and state relations so that the Church no longer seeks any establishment or acknowledgment as a true Church from the state (Paul VI 1965a, p. 1). The Church needs only freedom to act and not state authority. Civil powers have become incompetent in matters of religion. This marked a sharp break with the tradition of Church establishment, dating back to the times of Emperor Constantine (Burtchaell 1991).

## 3. The Perspectives of Contemporary Models of Secularism on the Catholic Church's Formula of Adaptation to Modernity

### 3.1. Contemporary Models of Secularism

This paper will employ Jacques Berlinerblau's modified definition of Secularism. The concept consists of political ideas, as well as legally binding actions of the secular state, that seek to regulate the relationship between the secular state and religion, especially between norms of the state and of religion, between the state and religious citizens, and between religious citizens themselves (Berlinerblau 2022, p. 5). Secular states are, generally, those in which there exists some sort of differentiation and separation between the state and religion.

Berlinerblau, a contemporary Secularism researcher, also offered a typology of models of Secularism that will be relevant to the purpose of this paper. His distinction was based on different states' attitudes and expectations towards religion, its norms, and its followers; it differed little from typologies used in contemporary scholarship on Secularism (Saeed 2017). The typology did not consider the modernization of the Catholic Church and her proposals on church–state relations; the aim of the present paper is precisely to look at these questions from the perspectives of the models. Five of them can be outlined as follows.

From the perspective of **Separationism**, all religious organizations (including churches) are or should be separate from the state and its institutions. The sphere of religion and its authorities cannot interfere with the sphere of the state and its institutions. In this framework, the aims and interests of religion and those of the state are considered to be basically differentiated. Although Jonathan Fox (2017) claimed that Separationism has a negative attitude towards religion, in this paper, the model will be treated on its own formal terms: the state neither restricts religion nor supports it. The state treats religious organizations, even the largest and most influential, in the same manner as private organizations—not granting them any privileges and not supporting any religious beliefs. Under this model, the freedom of individuals to choose (or not) their religious convictions is regarded as a crucial value. For some scholars and citizens, the model of Separationism has existed in its clearest form in the United States (Berlinerblau et al. 2014; Berlinerblau 2022).

**Disestablishment** is also a form of separation of religion from the state, but not as rigid as in Separationism. The main idea of Disestablishment is the lack of any established religion: there is no state religion, and no constitutional rule to support any particular religion. However, in contrast to Separationism, state support for religious organizations is not forbidden if—and only if—they perform activities valuable for the state or society. Examples might include social activities such as education or helping the poor. In this manner, religious organizations, including churches, are treated the same as non-religious organizations that have some valuable social aims. For many scholars and citizens, Disestablishment clearly exists in the United States and is expressed in the First Amendment (Berlinerblau et al. 2014; Berlinerblau 2022). There are, therefore, different opinions as to the model of the United States—whether it is, in fact, based on Separationism or Disestablishment. However, discussing this complex issue is not within the scope of this paper (see Nussbaum 2008; Feldman 2005). Certainly, both Separationism and Disestablishment are aimed at protecting freedom of exercise of religion, speech, the press, and assembly.

The perspective of **Laïcité** is of French origin and is said to have been aimed at eliminating religious conflicts by impeding the public dimension of religion and subordinating religion—to some extent—to state supremacy. It, therefore, contrasts with the two previous "Anglo-Saxon" models, which kept such subordination at a minimal level, with state interference only permissible in the case of visible disturbance to the public peace. State supremacy under Laïcité means, in practice, for example, prohibition of wearing religious symbols and attires in public, and confiscation of religious properties (as in France in 1905). However, after 1905 the French state started to subsidize the upkeep of religious buildings, and it continues to do so, especially in the case of Catholic churches; the state also finances the work of military chaplains. The Laïcité model seems partly hostile or at least suspicious towards religion, either at its root or as a consequence of state supremacy in many religious issues (Saunders 2009; Barras 2017).

Unlike Laïcité, **Accommodationism** believes religion to be a public good that should be supported by the state. It does not equate to establishment of any religion: Accommodationism respects Disestablishment, but it is far from Separationism. The focus on religion as a public good allows, for example, the acceptance of religious laws for particular religious groups (e.g., Sharia laws in India), even when they differ from the state's code of law. Accommodationism is likely to maintain public peace by influencing religion; however, its influence comes not through limiting public expressions of religion but by mediating and making public policies that encourage religious organizations to respect each other and contribute to the common good (Berlinerblau 2022).

In **Atheistic Secularism**, religion is regarded as a source of conflict and disruption. However, the proposed solution is much more radical than that of Laïcité: religion should not be limited but, in fact, eliminated from both public and private life. The aim of this model is not freedom of religion but rather freedom from religion. For this purpose, the atheistic worldview is favored and promoted by the state, and religion suffers severe handicaps, including property confiscation with no compensation, state influence over elections of religious leaders, prohibitions on confessing religion for persons holding public office, prohibitions on teaching religion in schools, and sometimes imprisonment or execution of clergy or religious people, as in Soviet Russia and contemporary North Korea (Berlinerblau 2022; Wanner 2012; Bandow 2019). The state has an official agenda of education, promoting non-religious worldviews that are profoundly rationalistic, including materialism.

### 3.2. The Perspective of Separationism

Separationism, like all other models of Secularism, must react positively to the Church's agreement on autonomy of science from religion, as it allows science to be placed within the field of interests of the state or society. Separationism would not forbid religious organizations from conducting their own research.

As Separationism is preoccupied almost solely with the strict separation of state and religion, its main assessment of *aggiornamento* will be connected with this issue. Generally, it will react positively to the idea that the Catholic Church will not seek establishment or state recognition and is focused on fulfilling its religious mission on its own. Because acceptance of the plurality of worldviews by the Church seems to be also foundational for Separationism (disestablishment would not be possible without this idea), this aspect of turning to Modernity could scarcely sound better from the perspective of Separationism.

However, this perspective will remain indifferent as to pluralism within the Catholic Church and other aspects of the Church's adaptation to Modernity. From the Separationist view, it does not matter whether the Catholic Church is ruled by an absolute monarch, a council of bishops, or popular, democratic voting among all her members. Although Casanova and many other scholars link Secularism with democracy (Casanova 2009, p. 1051), the model of Separation does not seem to require democracy or popular sovereignty on the Church's side. Due to its rule of non-interference, Separationism would tolerate any kind of governance among churches or other religious organizations as long as

they do not actively seek establishment. In the same manner, it does not matter whether the Church uses natural law or human rights categories or whether there is a strong position of the clergy or the laity. These kinds of ideas are treated by Separationism as internal issues of religious organizations that are not subject to state regulations or assessment.

Separationism would generally allow public discussion on religious matters, including the presentation of Catholic moral standards that differ from state norms, but would not allow instruments of governments to be used to advance any particular religious norms. While the Catholic Church's modernized formula of church–state relations does not rule out the possibility of state support for Catholic schools and education, financing the teaching of Catholic religion at public schools, financing Catholic military chaplains, or supporting Catholic social activities or projects, from the perspective of Separationism these undertakings would be violations of the principle of separation and should be strictly forbidden (Fox 2017).

### 3.3. The Perspective of Disestablishment

Disestablishment, like Separationism, positively assesses the retreat by Vatican II from the earlier standard that "the Catholic religion should be held as the only religion of the State, to the exclusion of all other forms of worship" (Pius IX 1864, p. 77). This means that the core Disestablishment values—pluralism, freedom of speech, and exercise of religion—have been accepted by the Catholic Church. In addition, Disestablishment, unlike Separationism, would accept some forms of state support for the Catholic Church as long as the Catholic religion is not established or unambiguously favored by the state. For example, under this model, financial support for Catholic schools has been considered possible: in the US, the Supreme Court confirmed in 2022 ("Carson v. Makin") that states cannot exclude religious schools from public benefits, a ruling that was very well received by the US Conference of Bishops (however, earlier rulings were not always in this direction—see Niose 2017). In Disestablishment, such support can be made not because any worldview is favored by the state but rather because of the rule of equality (if other private schools are supported by the state, then religious ones should be supported as well). Unlike Separationism, this model would also allow financial support for social actions, such as helping the poor, organized by the Catholic Church.

Disestablishment would probably remain indifferent to Church government: her level of democracy, sovereignty of members, language categories, pluralism, diversity within the Church, and many other effects of the modernization of the Catholic Church. Like Separationism, Disestablishment would not interfere in the affairs of religious organizations, giving them as much freedom as possible.

However, one difference between Disestablishment and the Catholic Church's version of the separation of church and state should be noted. While disestablishment is a fundamental requirement for this model of Secularism, this does not seem so for the Catholic Church's formula. Vatican II's idea that the Church does not seek establishment does not mean that the state cannot seek it or that the Church must not accept it. The modern example of such a relation is the Constitution of Malta, Article 2, which declares: "The religion of Malta is the Roman Catholic Apostolic Religion," and "The authorities of the Roman Catholic Apostolic Church have the duty and the right to teach which principles are right and which are wrong" (Constitution of Malta 1964). The Catholic Church has not objected to this establishment. From the perspective of Disestablishment, however, it is obviously a violation of its fundamental principle.

### 3.4. The Perspective of Laïcité

Unlike the two previous perspectives, Laïcité's aim is not to grant the Catholic Church—or any other religious organization—as much freedom as possible, but rather to subordinate her to the state as much as seems necessary. One of the reasons is the genesis of this perspective. While Disestablishment, and to some extent Separationism, was created with an accompanying idea that Christian churches are valuable elements of the public order,

Laïcité, crafted by *les philosophes* who were critical of the Catholic Church, has always been suspicious about the influence of religion in civil society (Berlinerblau 2022, p. 66). As a result, Laïcité objects to the scope of the freedom proposed by the post-conciliar Catholic Church, and would certainly prefer to find in the modern Catholic formula an exposition of the role of the state in directing some religious issues. It favors the formula of non-establishment of the Church (accepted by the Church also), but not strict separation, since the state, according to this perspective, should have a considerable level of control over the Church. This is not in accordance with the Catholic formula of adaptation to Modernity. In particular, from the perspective of Laïcité, the Church's property should belong to the state; such a proposal cannot be found in the Catholic formula. Laïcité would agree with the Church's formula only in its second step—subsidizing of Church buildings by the state—and then only if they became state property. More generally, Laïcité removes religion from the public sphere (Bauberot 2014), or introduces a "Rigid Control Model" (Drury 2017, p. 11); no such idea is indicated in the Catholic Church's adaptation to Modernity.

Although Laïcité's influence on religion is not so extensive as to affect or propose changes to Church government, the Catholic Church's focus on individual, human rights and elements of democracy, including popular sovereignty, must be favorable from this perspective. Laïcité's partial control over religion is said to be justified by the need to eliminate religious conflicts and maintain public peace; the liberal instruments of human rights adopted by the Church serve this end also, by protecting individuality and democracy. Therefore, these aspects of the Catholic modern formula would be regarded positively from the Laïcité perspective, as they might indirectly support Laïcité's aims.

### 3.5. The Perspective of Accommodationism

Accommodationism must place high value on the Catholic Church's formula of separation of church and state, firstly because of disestablishment, but also because it regards religion as a public good. Unlike Laïcité, Accommodationism does not tend to confine religion to the private sphere. It seems to understand the Catholic Church's position that "religious mission can be the source of commitment, direction and vigor to establish and consolidate the community of men" (Paul VI 1965b, p. 42). Accommodationism, therefore, acknowledges that religion, including Catholicism, may contribute to the common good of society; it does not see a need to limit its public expressions, such as religious attires. This kind of passive Secularism (Kuru 2007) is both favored by Accommodationism and accepted by the Catholic Church.

As Accommodationism regards religion as a public good, it should not be indifferent towards pluralism and the human rights categories presented in the Catholic formula of adaptation to Modernity. Multicultural India, where the model is most clearly manifested, seeks ways to strengthen peace and might highly value these aspects of modern Catholicism's proposals. In addition, Accommodationism is not particularly committed to equal treatment of religions; it may give more public rights to one religion, not necessarily to the prevailing one. In this case, Accommodationism is sometimes referred to as "pseudo-secularism" as it tends to declare a secular state, at the same time "pampering" some religious minorities (Pantham 1997, p. 528). Anyway, although the Catholic Church does not seek special treatment from the state, she would probably accept or welcome Accommodationism's inequality if it took the form of public recognition of her laws, as it has in the case of Islam in India, or as Malta has done for Catholicism. The first scenario theoretically exists in Accommodationism; however, the second would be unacceptable to the model due to formal establishment, departing from the formula of neutrality (Berlinerblau 2022).

It is worth noting that although, theoretically, the model of Accommodationism seems to be very close to the Catholic Church's formula of adaptation to Modernity in crucial points, the perspective of the model does not always align with the practical dimension. This is especially so with social attitudes, as religiously motivated violence against Christians in India has not been a rare phenomenon in recent years (Human Rights

Watch 2009). In addition, inequality under Hindu Accommodationism has tended to favor Hinduism and Islam, while twenty-first-century anti-conversion laws in a number of Hindu states are said to be easily used to sentence Christians, regardless of whether they have attempted conversions of others (Gettleman and Raj 2021). If this practical inequality is treated as an element of Accommodationism, then it would not regard positively the Catholic Church's modern call for religious freedom.

### 3.6. The Perspective of Atheistic Secularism

The last perspective would assess the Catholic Church's formula of adaptation to Modernity decisively negatively. This is not merely because Atheistic Secularism contests reforms towards Modernity, but also because it contests religion, including Catholicism, regardless of whether it is an older or adapted version. Atheistic Secularism cannot accept either Catholic establishment—as promoted strongly before Vatican II—or a Catholic version of separation of church and state. In the former case, the perspective cannot agree to any public role for Catholicism, as it desires for the atheistic worldview to be publicly promoted in a state's Constitution, in its institutions, and in education. Instead of establishing Catholicism or any other religion, it establishes Marxism or another non-religious worldview. The latter case looks better for Atheistic Secularism, but it is still not favored, as the Catholic version of separation still gives the Church a relatively large degree of freedom to act and influence the public and private spheres. That is the exact opposite of what the atheistic perspective would like to achieve. It would favor the independence of science and religion, and would either disregard or not allow science to be cultivated inside religious organizations. Moreover, Atheistic Secularism, unlike the Catholic Church's formula of adaptation to Modernity, would not accept pluralism of religion as it aims to eliminate religion from public and private life (Berlinerblau et al. 2014; Wanner 2012).

The Catholic *aggiornamento*, introducing elements of democracy into the Church, altering its government, and focusing on human rights, is not of particular interest for Atheistic Secularism, which would remain indifferent to such matters. The model would welcome only a version of Church government that would allow the state to interfere in Church affairs, to keep the Church under state control and hinder religious influence. Such a model, thwarting freedom of religion, is, of course, unacceptable for Modern Catholicism, which promotes freedom of religion and accepts pluralism (Ngo and Quijada 2015).

Generally, the perspective of Atheistic Secularism, at least in the version outlined above, denies the most crucial ideas of Modernity, especially those embraced by the Catholic Church. It is not preoccupied with individualism, human dignity, freedom of religion, or pluralism. Instead, it promotes a non-religious worldview and very often (in Soviet Russia and most Communist states) class interests, to which it subjugates individuality and human dignity. In North Korea a political religion called Juche, with a cult of personality around the atheistic leader, has been promoted for decades, which has resulted in officially allowing to function only one, state-subjugated, Catholic church parish in Pjongjang (Bandow 2019). For the Catholic Church, it must resemble the situation of ancient Roman Empire with the cult of emperors and Christian persecutions.

### 4. Conclusions

The Catholic Church made its own way to Modernity, adopting many modern ideas at Vatican Council II and afterwards. The most significant of these are the separation of church and state, religious freedom and pluralism, individual human dignity and human rights, elements of democracy, and the "independence of worldly affairs" from religion (Paul VI 1965b, p. 36).

The Catholic Church's formula for adaptation to Modernity can be assessed differently depending on one's perspective. In this paper, Jacques Berlinerblau's typology of models of Secularity (Berlinerblau 2022) provided five perspectives to assess the Church's modernization. These perspectives are summarized below, beginning with the most and ending with the least positive assessment.

Looking at the theoretical assumptions of the models, the most positive assessment of the Catholic Church's formula for adaptation to Modernity would be given by Accommodationism. This model would particularly appreciate the Catholic idea of church and state separation. The Catholic insistence on treating religion as a public good would enable the state to support religion, to some extent, in order to benefit society. However, practical applications of the model—see the example of India—seem rather problematic, as does treating Catholicism as a public good, as justified in the paper.

A similarly positive assessment of the Catholic model is given by Disestablishment. This model also appreciates the Catholic idea of separation of church and state, and the possibility for a religious organization to be supported by the state, in some cases, without an established religion. However, the model would not support establishment of religion in any state; in practice, the Catholic Church has not fully abandoned this model of church and state relations, as in the case of Malta.

The third most positive assessment of the Catholic formula for adaptation to Modernity is given by Separationism. This view would positively assess the Catholic version of separation of church and state and the Church's need for freedom to act. On the other hand, the model prefers stricter separation than proposed by the Church—in particular, it would not agree to subsidize any public mission of the Church, including social or educational actions.

A slightly less positive assessment would be made by the Laïcité model, which is committed to limiting some aspects of the public mission of the Church, and prefers to submit religion to state supremacy, at least to some extent. This model is, therefore, suspicious of the Catholic ideas of freedom of religion and the Church's wide range of freedom to act; however, it would still agree to a certain level of freedom.

The only model to regard the Catholic Church's formula for adaptation to Modernity decisively negatively is Atheistic Secularism (at least in the Soviet version). Firstly, it generally regards religion as problematic for the state and society; secondly, it does not favor the Catholic version of separation of church and state. Instead, it would prefer stricter submission of the Church to the state in order to eliminate or severely weaken the influence of religion on society (Ngo and Quijada 2015).

**Funding:** This research received no external funding.

**Institutional Review Board Statement:** Not applicable.

**Informed Consent Statement:** Not applicable.

**Data Availability Statement:** Any data cited has been previously published and included in the references.

**Conflicts of Interest:** The author declares no conflict of interest.

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
