# Peer review of "The Catholic Church’s Formula for Adaptation to Modernity and Contemporary Models of Secularism"

_religions, doi:10.3390/rel14050638_

Round 1
Reviewer 1 Report
The article carries out an interesting exercise of analysis of the adaptation of Catholicism to modernity based on the different proposals defined by Berlinerblau. We would like to congratulate the authors for their proposal. Having said this, we would like to point out a number of issues that may help to improve the impact of the article.
-The characteristics of modernity defined by the authors are linked to a specific cultural program, the Judeo-Christian one. In this sense, and in order to qualify the categorical nature of the statement, it is perhaps interesting that the authors go deeper into Eisenstadt's concept of multiple modernities (which they cite in the text). Perhaps this is the major shortcoming of the paper, since when the authors talk about the application of Berlinerblau's model to different specific contexts they do not fall back on the specificities of the different cultural programs associated with modernity. It would be interesting if this aspect of the paper could be reviewed.
-Very interesting is the distinction made by the authors between aspects in which the church has adapted to modernity and others in which it has not. We think is an idea that can be developed more.
-If the authors read in Spanish, perhaps it would be interesting for them to read Javier Gil-Gimeno's work "De la confrontación al aggiornamento en las relaciones entre iglesia católica y modernidad" (From Confrontation to Aggiornamento in the Relationship between Catholic Church and Modernity), published in the Mexican journal Sociológica.
-Berlinerblau's work is not cited in the final bibliography.
Congratulations to the authors.
Author Response
Dear Reviewer, thank you for your comments and care to improve the text. Thanks to it made the following modifications to it
- In order to take into account Eisenstadt’s idea of Mutliple Modernities more thoroughly, the section defining Modernity (2.1) has been modified in the manuscrpit– one paragraph and more specific qualifications have been added. Especially, the definition of Modernity presented in the paper has been confined to the “Western” understanding of Modernity and justifications of such definition – for the purpose of the paper – are now presented. Also, Eisenstadt’s work is now referred to and added to bibliography. Hopefully it makes definitions more clear and the general statement less categorical.
2. The section on aspects in which the Church did not adopt to Modernity has been enlarged – a few sentences showing instances of this non-adoptation were added (lines 112-120).
3. Unfortunately, the author does not speak Spanish and is unable to read the suggested position at that moment, but the author is encouraged to deepen his knowledge in mentioned aspects.
4. Berlinerblau’s works are now cited in bibliography.
Reviewer 2 Report
The article is short and clear. One of the objections could be the simplification of the phenomenon of Modernity, but I suppose that the author made that decision in order not to burden the text and spread the discussion with more or less generally accepted assumptions.
Some observed errors:
Line 4: The aim of this paper (not his paper)
References: The “main” source, i.e. Berlinerblau, is not listed in the references.
Line 470: Reference is not complete. Missing source.
Author Response
1. Simplification of the phenomenon of Modernity is now shortly discussed and justified (additional paragraph in section 2.1). Yes, the aim of simplification was to focus on other issues and not on general modernity discussion. However, Modernity seems now a bit more specifically defined – in the “Western” pattern.
2. Indicated errors in lines have been corrected. The source has now full data. Berlinerblau’s works are now presented in bibliography.
Reviewer 3 Report
The researcher has appropriately dealt with the research questions raised in para 3. The Catholic formula has been analyzed based on Berlinerblau's typology.
Secularism in India was discussed under Accomodationism. Lines 322-23 talk about the possibility of giving "more public rights to one religion". It might help if the writer can mention the issue of "pseudo-secularism" raised by some in this context.
It might be interesting to see what the researcher thinks about the type of secularism practiced in North Korea and how the Catholic Church relates to it.
Overall, the paper contributes well to the discussion of the current state of secularism in the world.
Author Response
1. The issue of pseudo-secularism in now mentioned in the text when speaking about Indian Accommodationism (lines 362-365) and seems indeed very relevant in this place. One source on the topic is indicated and added to bibliography.
2. After short research, North Korea has been added and shortly explained as an example of State-Sponsored Atheism Model. One source on the topic is indicated and added to bibliography. Thank you for pointing out to this interesting, contemporary case.